# Persister Cell Formation and Elevated *lsrA* and *lsrC* Gene Expression upon Hydrogen Peroxide Exposure in a Periodontal Pathogen *Aggregatibacter actinomycetemcomitans*

**DOI:** 10.3390/microorganisms11061402

**Published:** 2023-05-26

**Authors:** Yohei Nakamura, Koji Watanabe, Yoshie Yoshioka, Wataru Ariyoshi, Ryota Yamasaki

**Affiliations:** 1Division of Infections and Molecular Biology, Department of Health Promotion, Kyushu Dental University, Kitakyushu 803-8580, Fukuoka, Japan; r19nakamura@fa.kyu-dent.ac.jp (Y.N.); r16yoshioka@fa.kyu-dent.ac.jp (Y.Y.); arikichi@kyu-dent.ac.jp (W.A.); 2Division of Developmental Stomatognathic Function Science, Department of Health Promotion, Kyushu Dental University, Kitakyushu 803-8580, Fukuoka, Japan; r17watanabe2@fa.kyu-dent.ac.jp; 3Collaborative Research Centre for Green Materials on Environmental Technology, Kyushu Institute of Technology, 1-1 Sensui-chou, Tobata-ku, Kitakyushu 804-8550, Fukuoka, Japan

**Keywords:** *Aggregatibacter actinomycetemcomitans*, hydrogen peroxide, persister, periodontitis

## Abstract

The effect of hydrogen peroxide, an antiseptic dental treatment, on *Aggregatibacter actinomycetemcomitans*, the main causative agent of localized invasive periodontitis, was investigated. Hydrogen peroxide treatment (0.06%, 4× minimum inhibitory concentration) resulted in the persistence and survival of approximately 0.5% of the bacterial population. The surviving bacteria did not genetically acquire hydrogen peroxide resistance but exhibited a known persister behavior. Sterilization with mitomycin C significantly reduced the number of *A. actinomycetemcomitans* persister survivors. RNA sequencing of hydrogen peroxide-treated *A. actinomycetemcomitans* showed elevated expression of Lsr family members, suggesting a strong involvement of autoinducer uptake. In this study, we found a risk of *A. actinomycetemcomitans* persister residual from hydrogen peroxide treatment and hypothesized associated genetic mechanisms of persister from RNA sequencing.

## 1. Introduction

*Aggregatibacter actinomycetemcomitans* is a bacterium associated with invasive periodontitis. It is particularly prevalent in areas affected by rapidly progressing periodontitis, previously classified as aggressive periodontitis, which is induced by the ability to form biofilms in the subgingival marginal space [1,2]. Earlier, the term “juvenile periodontitis” was used to describe the disease, which primarily affected younger patients below 35 years of age, but the case definition has shown that the disease is associated with a variety of age groups [3]. Seven serotypes of *A. actinomycetemcomitans* have been identified based on surface carbohydrate antigens [4]. Of these, serotypes a, b, and c predominate worldwide, with serotype c being the most prevalent and serotype b being most frequently associated with periodontitis [5]. Serotype b *A. actinomycetemcomitans* Y4 strain was used in this study. This strain was isolated from a patient with localized invasive periodontitis at the Forsyth Dental Institute (Boston, MA, USA) in 1979 and has been used for many years as a model strain for laboratory studies [6]. *A. actinomycetemcomitans* is associated with various diseases besides oral infections, such as bacteremia, sepsis [7,8], endocarditis [8,9], atherosclerosis [10], pneumonia [11], Alzheimer’s disease [12], skin infections, osteomyelitis, infectious arthritis [13], diabetes [14], urinary tract infections [15,16], and various types of abscesses [17,18,19]. Whether systemic translocation through the epithelial barrier is due to an active invasive process or passive leakage into the bloodstream is unknown; however, the disease can be prevented by sterilizing the *A. actinomycetemcomitans* with drug treatment in the oral cavity [19]. Various drugs and/or antiseptics are commonly used for dental treatment, but the concept of “persister” is an important factor to be considered when administering them.

A persister is a resistant bacterial phenotype formed upon exposure to extreme environmental conditions, such as drugs or starvation [20,21,22]. It differs from drug-resistant mutants because the portion of the bacterial population that survives drugs and other stresses is not genetically mutated [23]. Since its first recognition by Hobby et al. in 1942 [24], many studies have elucidated the mechanism of persisters [25,26,27,28]. Although genetic drug resistance due to mutations is the primary cause of difficulties in drug therapy of bacterial infections, the presence of a persister may also have a significant role. In particular, the persister may be the main cause of repeated treatment and relapses with the same drug, as the use of the same drug for the treatment of drug-resistant bacteria after recurrence is not effective. This is also applicable to oral diseases, which are mainly treated with drugs for dental treatment. Therefore, it is presumed that persistent periodontal disease (chronic periodontitis) and other oral diseases are difficult to cure completely because persisters survive drug treatment and remain in the affected area, where they can re-grow.

Hydrogen peroxide is a commonly used dental agent. Apart from dentistry, hydrogen peroxide is widely used for bleaching and deodorization of industrial products and food, treating sewage, disinfection, and manufacturing many chemicals and chemical products [29,30,31,32]. It also has broad-spectrum antimicrobial activity, with activity against bacteria, fungi, viruses, protozoa, and prions [33]. The bactericidal effect of hydrogen peroxide is mainly due to the oxidizing power of the reactive oxygen species produced [34], which penetrate the cell membrane and act internally to kill bacteria [35]. In addition, hydrogen peroxide does not emit toxic substances and has a low impact on the human body and the environment at commonly used concentrations [36]. Therefore, it is used in various fields, including dentistry. However, there is a risk of persister survival even when hydrogen peroxide is used as an oral disinfectant. Therefore, this study aimed to verify whether persisters remain when hydrogen peroxide used in dental treatment is applied to *A. actinomycetemcomitans*. We also aimed to elucidate the rate and mechanism of persistence and identify a method for its complete elimination. Proving persister survival and clarifying the mechanism of persister formation will provide new concepts and treatment methods in dentistry in the future.

## 2. Materials and Methods

### 2.1. Cultivation of A. actinomycetemcomitans Y4 and Minimum Inhibitory Concentration of Hydrogen Peroxide

*A. actinomycetemcomitans* Y4 was used as the model strain in this study. *A. actinomycetemcomitans* Y4 was streaked from the glycerol stock onto brain–heart infusion (BD, Franklin Lakes, NJ, USA) containing 1% yeast extract (BD, Franklin Lakes, NJ, USA) (BHIY) agar and incubated at 37 °C and 5% CO_2_ for two days. A single colony was inoculated into BHIY broth and incubated at 37 °C and 5% CO_2_ overnight. Next, BHIY containing 0.96% *v*/*v* hydrogen peroxide (FUJIFILM Wako Pure Chemical Corporation, Osaka, Japan) was prepared to achieve different concentrations of hydrogen peroxide (0.96–0% *v*/*v*) and diluted 2-fold onto a 96-well microtiter plate (Iwaki, AGC Techno Glass Co., Ltd., Shizuoka, Japan). The overnight culture was inoculated at an optical density of 0.05 at 600 nm into each well of the 96-well plate and incubated at 37 °C and 5% CO_2_ for 24 h. Growth inhibition was measured at 620 nm using a microplate reader (Thermo Fisher Scientific K.K., Tokyo, Japan), and the completely inhibited (significantly inhibited) concentration was determined as the minimum inhibitory concentration (MIC). All experiments were performed using at least three biological replicates.

### 2.2. Bactericidal Effect and Genetically Antiseptic Resistant Confirmation of Hydrogen Peroxide against A. actinomycetemcomitans Y4

*A. actinomycetemcomitans* Y4 overnight culture was re-inoculated into fresh BHIY broth and incubated at 37 °C and 5% CO_2_ to an optical density of 0.4 at 600 nm. The culture was centrifuged at 3500× *g* for 10 min and washed twice with 1× phosphate-buffered saline (PBS). The bacterial pellet was resuspended in PBS containing 4× MIC hydrogen peroxide and incubated at 37 °C and 5% CO_2_ for 0.5, 3, 6, 12, and 24 h. After each incubation, the culture was 10-fold serial diluted using PBS and spot-plated on BHIY agar to determine the number of colony-forming units (CFUs). Bacterial suspension treated with 4× MIC hydrogen peroxide for 3 h was centrifuged at 3500× *g* for 10 min, washed twice with PBS, resuspended in BHI, and incubated at 37 °C and 5% CO_2_ for 15 h to confirm whether the bacteria surviving after hydrogen peroxide treatment were not genetically antiseptic resistant. The bacterial culture was again treated with 4× MIC of hydrogen peroxide to quantify the reduction in bacterial abundance. The hydrogen peroxide concentration during the bactericidal effect test was confirmed using MONITOR^TM^ for HYDROGEN PEROXIDE 0–0.04% (Serim Research, Elkhart, IN, USA).

### 2.3. Persister Cells Resuscitation Time on Agarose Gel Pads

Agarose gel pads were prepared for microscopic observation using Kim et al.’s method [37]. Briefly, 1.5% agarose (NIPPON GENE, Tokyo, Japan) was added to BHIY broth and melted by microwaving (150 sec at 500 W). Melted BHIY-agarose was poured into the slide glass template and raised to solidify. Exponential state (OD_600_ = 0.4) and hydrogen peroxide-treated (3 h) bacterial culture of *A. actinomycetemcomitans* Y4 were centrifuged at 3500× *g* for 10 min and washed with PBS, respectively. Further, 10 µL of each was placed on the gel pads and observed with a cover glass at 1000× magnification under a microscope (BZ-X 800; KEYENCE CORPORATION, Osaka, Japan). The environmental conditions during microscopy were maintained at humidity, 37 °C, and 5% CO_2_ using a temperature and CO_2_ control chamber. All the analysis points were selected randomly, avoiding areas of bacterial aggregation. The number of bacteria cell divisions was counted every 30 min.

### 2.4. Sterilization of A. actinomycetemcomitans Y4 Persister Using Mitomycin C

*A. actinomycetemcomitans* Y4 was treated with 4× MIC hydrogen peroxide for 6 h. Survived persister cells were centrifuged at 3500× *g* for 10 min and washed twice with PBS. The bacterial pellet was resuspended in PBS containing 10× MIC mitomycin C (MMC) (1.25 µg/mL) and incubated at 37 °C and 5% CO_2_. After MMC treatment, the culture was centrifuged at 3500× *g* for 10 min and washed twice with PBS to remove residual MMC; it was then 10× serially diluted and spot-plated on BHIY agar at 0.5, 3, 6, 12, and 24 h to determine the number of CFUs.

### 2.5. Transcriptome Analysis of A. actinomycetemcomitans Y4 Using RNA Sequencing

Hydrogen peroxide-nontreated *A. actinomycetemcomitans* Y4 and -treated persister *A. actinomycetemcomitans* Y4 were prepared at 1 × 10^7^ CFUs. Total RNA isolation was performed using the RNeasy mini kit (QIAGEN, Fenlo, Netherlands) according to the manufacturer’s instructions. Briefly, bacterial cell disintegration was performed using bashing beads (ZYMO, Irvine, CA, USA), and DNase treatment was performed to remove residual genomic DNA from the RNA sample. Extracted RNA was reverse transcribed into complementary DNA using the ReverTra Ace^®^ qPCR RT Master Mix (TOYOBO, Osaka, Japan). RNA sequencing was performed using Nippon Genetics (Tokyo, Japan). In addition, the expression levels of several important genes obtained using RNA sequencing were verified using quantitative real-time reverse transcription-polymerase chain reaction (RT-PCR) (Agilent Technologies, Santa Clara, CA, USA). Reactions were prepared using Brilliant III Ultra-Fast SYBR^®^ Green qPCR Master Mix with LOW ROX (Agilent Technologies) and an AriaMX Real-Time PCR system using the following primer sequences: *adk*, 5′-ACCGGCGATATGTTACGTTC-3′ (forward) and 5′-ATTCCTTGCTCACAGCTTCC-3′ (reverse); *lsrA*, 5′-GAGCCAAAATGCTTAATATCCGCC-3′ (forward) and 5′-TCAAATCCTGCACCTGCAAAATCG-3′ (reverse); *lsrC*, 5′-ACGGCTTTCATCTGCAAACGTTAA′ (forward) and 5′-CGATACCGGCAACAAAATTGTTCC-3′ (reverse). Relative changes in gene expression were calculated using the comparative threshold cycle. Total cDNA abundance between samples was normalized using primers specific to *adk*.

## 3. Results

### 3.1. MIC of Hydrogen Peroxide for A. actinomycetemcomitans

The growth inhibition of *A. actinomycetemcomitans* in the hydrogen peroxide-added group was determined at an absorbance of 620 nm relative to that in the group without hydrogen peroxide (Figure 1). Hydrogen peroxide inhibited the growth of *A. actinomycetemcomitans* at concentrations ≥0.015%. Therefore, 0.015% hydrogen peroxide was determined as the MIC.

### 3.2. Bactericidal Effect of Hydrogen Peroxide on A. actinomycetemcomitans and the Confirmation of Persistence

The bactericidal effect of *A. actinomycetemcomitans* was verified using 4× MIC of hydrogen peroxide and spot-plating the bacterial cultures at 0, 0.5, 3, 6, 12, and 24 h. The results showed that approximately 99.5% of *A. actinomycetemcomitans* were killed at 3 h, following which the bacterial count stabilized until 24 h (Figure 2a,b). These bacteria probably evaded the action of hydrogen peroxide and were considered persisters. Because the *A. actinomycetemcomitans* cells possibly acquired antiseptic resistance due to genetic mutations, the re-grown cultures were treated with the same concentration of hydrogen peroxide. *A. actinomycetemcomitans* re-grew on incubation in hydrogen peroxide-free BHIY medium after 3 h of hydrogen peroxide treatment. The culture was saturated after 16 h of incubation. Treatment with 4× MIC hydrogen peroxide again reduced the bacterial count to the pre-treatment level (Figure 2c). Therefore, the cells that survived hydrogen peroxide treatment were persisters and not genetic mutants. Since *A. actinomycetemcomitans* cells may have survived because of reduced hydrogen peroxide concentration, the concentration of hydrogen peroxide in the *A. actinomycetemcomitans* culture medium during treatment was examined using a test paper. In the experiment shown in Figure 2b, the hydrogen peroxide concentration in the *A. actinomycetemcomitans* culture was determined at 0, 0.5, 12, and 24 h, and no decrease in concentration was observed (Appendix A). Bacterial culture with 0.06% hydrogen peroxide showed concentrations ≥0.06% at 0 h, and the color of the test paper did not change after 24 h (Appendix A). Therefore, the cells survived because they were persisters and not because of a decrease in hydrogen peroxide concentration.

### 3.3. Resuscitation Time of A. actinomycetemcomitans Persister Cells

The resuscitation (division) time of persister cells is widely divergent compared to that of exponential cells [37]. Therefore, the division start time of *A. actinomycetemcomitans* exponential or hydrogen peroxide-treated cells (persister cells) on BHIY gel pads was investigated using a microscope. To ensure sufficient data, a total of 101 exponential cells and 4291 persister cells were observed. Figure 2b shows that hydrogen peroxide treatment for 3 h killed most of the bacteria, and a very small number (approximately 0.5%) developed into persisters. Therefore, higher bacterial counts were observed than those in the exponential cells. Ninety-eight of 101 exponential cells (97.03%) and 49 of 4291 persister cells (1.14%) divided. The division time of each group is indicated in Figure 3 and Appendix A. The division of exponential cells initiated from 1.5 to 3 h, and more than 90% of the cells divided between 1.5 and 2 h. In contrast, the division initiation times of persister cells differed greatly; the early cells initiated division at 1.5 h and the late cells at 6 h. Between 1.5 and 6 h, approximately 0.1% of the cells were resuscitated (Appendix A).

### 3.4. Mitomycin C Kills Persister Cells

In the experiments described above, a small fraction of *A. actinomycetemcomitans* cells survived by developing into persisters to hydrogen peroxide. The next issue was to eliminate the persisters. The anti-cancer drug MMC has an efficient sterilization effect against persister cells [38]. Here, the sterilization of *A. actinomycetemcomitans* persister cells was confirmed at 10× MIC of MMC (1.25 µg/mL). The MIC data of MMC against *A. actinomycetemcomitans* are shown in Appendix A. As a result, zero viable counts were achieved after more than 3 h of treatment (Figure 4), demonstrating the efficacy of MMC against *A. actinomycetemcomitans* persisters.

### 3.5. RNA Sequencing of Gene Expression Levels after Hydrogen Peroxide Treatment

The differences in gene expression in hydrogen peroxide-treated and untreated *A. actinomycetemcomitans* cells were confirmed using RNA sequencing. The results showed that the expression levels of *lsrA* and *lsrC* in hydrogen peroxide-treated *A. actinomycetemcomitans* were significantly higher, 2.51-fold and 5.28-fold, respectively, than those in non-treated cells (Table 1), suggesting the importance of these genes in surviving the hydrogen peroxide environment. The expression of *lsrR*, which suppresses the expression of the Lsr family, decreased 0.75-fold. Autoinducer-2 (AI-2) is taken up by the Lsr family, and the expression of its synthase, *luxS*, remained virtually unchanged at 0.95-fold. Gene expression levels of *lsrA* and *lsrC* were further confirmed using real-time RT-PCR. The mRNA levels of *lsrA* and *lsrC* were 2.2 ± 0.7-fold and 8.5 ± 2.6-fold higher, respectively, in hydrogen peroxide-treated persisters, compared to those in the non-treated control (Appendix A).

## 4. Discussion

### 4.1. The Concept of “Persister” Is Important in the Treatment of Periodontal Disease

Due to the high prevalence of rapidly progressing periodontitis in people below 35 years of age before 2007, it had been termed juvenile periodontitis [39]. However, periodontitis is a disease relevant to all age groups. This study demonstrates the bactericidal effect of hydrogen peroxide, a dental agent, against the causative agent of localized invasive periodontitis, *A. actinomycetemcomitans*, and its associated challenges. Periodontopathogenic bacteria, such as *A. actinomycetemcomitans*, form biofilms in the gingival sulcus (the groove between the teeth and gingiva), triggering gingival inflammation and periodontitis [40]. In such conditions, drugs or antiseptics used for dental treatment and oral care may be ineffective (due to extremely low concentrations). The concentration of hydrogen peroxide used in general dental treatment and oral care is 0.75–30% [36]; however, the working concentration of hydrogen peroxide is considered to be much lower than this range due to the dilution effect of saliva. Our results showed that 4× MIC (0.6%) hydrogen peroxide formed persisters against *A. actinomycetemcomitans* (Figure 2b). Thus, the presence of genetically stable oral pathogenic bacteria surviving hydrogen peroxide treatment is evident. Importantly, periodontal disease may recur or become chronic because the bacteria survive antiseptic treatment as persisters. Since Figure 2c shows that bacteria thriving after hydrogen peroxide treatment are not mutants, we must acknowledge the concept of “persisters” in addition to bacteria that have acquired genetic drug resistance. To corroborate that hydrogen peroxide-treated *A. actinomycetemcomitans* form persisters, we measured the time required for resuscitation. Persister cells resuscitate and initiate division or elongation when exposed to a carbon source or other nutrients, but they exhibit greater variation in division initiation time than that observed with exponentially growing bacteria [37]. The resuscitation time of *A. actinomycetemcomitans* persister cells surviving hydrogen peroxide treatment also varied widely, ranging from 1.5 to 6 h, compared to almost 1.5 to 2 h for the exponential cells (Figure 3).

Eliminating all persister cells completely cures the infection. The anti-cancer drug MMC can sterilize *Escherichia coli* persister cells [38]. MMC functions by inhibiting DNA replication via the prevention of DNA division and DNA strand breaks caused by reactive oxygen species [41]. MMC is passively transported inside bacterium and is bioreductively activated, causing spontaneous cross-linking of DNA; MMC activity does not require active metabolism, making it effective against persister cells [38]. The use of MMC on *A. actinomycetemcomitans* persister cells surviving hydrogen peroxide treatment resulted in immediate and complete sterilization within 3 h (Figure 4). Therefore, MMC was effective against *A. actinomycetemcomitans* persisters. However, since MMC is a potent antibiotic, it is necessary to carefully examine its safety, drug resistance, and applicability to the oral cavity prior to administration.

### 4.2. Inference of Persister Formation Mechanism from Gene Expression Levels

Bacteria communicate among species using quorum sensing. Bacteria that engage in quorum sensing produce intracellular autoinducers; *A. actinomycetemcomitans* use AI-2 (4,5,-dihydroxy-2,3-pentanedione) as an intercellular signaling molecule. AI-2 is involved in persister formation; a decrease in AI-2 levels decreases the abundance of persister cells, increasing susceptibility to death [42,43,44]. The Lsr family is a transporter for the uptake of AI-2. In the present study, gene expression of *lsrA* and *lsrC* in *A. actinomycetemcomitans* greatly increased by hydrogen peroxide treatment, while that of *lsrR*, a repressed gene of the Lsr family, decreased. Hence, the increased expression of the Lsr transporter may have increased the uptake of AI-2 and escaped the bactericidal effect of hydrogen peroxide by persister formation. Moreover, deletion of the AI-2-producing gene in *Salmonella* ser. Typhimurium decreases its ability to produce catalase in acidic environments and increases susceptibility to bile [44]. Therefore, the quorum sensing system of *S.* Typhimurium may help manage oxidative stress, and increased persister cell populations could aid in the chronic persistence of the bacteria in the gallbladder. In our study, catalase gene expression in *A. actinomycetemcomitans* increased 2.52-fold after hydrogen peroxide treatment (Table 1). Therefore, the uptake of AI-2 by the Lsr family may be involved in the upregulation of catalase expression, and the catalase produced may reduce oxidative stress. In addition, the involvement of glycerol in regulating the Lsr operon has been reported [45]. Phosphorylated glycerol (glycerol 3-phosphate) suppresses cAMP-CRP and inhibits Lsr operon activation. Glycerol is an important resource used in the metabolism and fermentation of various organisms. Since hydrogen peroxide inactivates the ribosomal translational activity [46], and the bacteria that form persisters have stopped metabolic activity [37], it can be assumed that glycerol fermentation is suppressed. Therefore, *A. actinomycetemcomitans* cells that form persisters upon hydrogen peroxide treatment do not produce glycerol 3-phosphate, and cAMP-CRP is not inhibited and remains normally active, thus activating the Lsr operon. As a result, they survive, forming persisters to hydrogen peroxide stress, leading to AI-2 uptake. In addition, cAMP-CRP is greatly involved in ribosome activity in persisters and suppresses the ribosome rescue factor Hfq (the same operon as HflX), which induces ribosome dormancy, i.e., persister [28]. Moreover, ribosomal rescue via the translational system (SsrA) is reportedly important in the resuscitation of persisters [47], and the expression levels of these two genes (*hfq* and *ssrA*) were suppressed in hydrogen peroxide-treated *A. actinomycetemcomitans*, with *hfq* being suppressed by 0.91-fold and *ssrA* by 0.31-fold (Table 1). The induction of various persister mechanisms, such as the involvement of autoinducers, avoidance of oxidative stress by catalase, and even deactivation of ribosomal activity, result in the survival of hydrogen peroxide treatment. These considerations can be summarized as follows: 1. Hydrogen peroxide stress increases Lsr family expression and AI-2 uptake; AI-2 suppresses LsrR, a repressor of Lsr family, and thus promotes further Lsr family expression; 2. Increased AI-2 uptake increases catalase production, scavenging reactive oxygen species generated from hydrogen peroxide; and 3. the reduction in metabolic activity arrests the glycolytic system; thus, glycerol 3-phosphate is not produced and cAMP-CRP activity is not suppressed. This leads to further induction of Lsr family expression and suppression of Hfq, a ribosomal rescue factor. It also suppresses SsrA, another ribosome rescue factor whose function is unknown, resulting in ribosome inactivation (Figure 5). We speculate that *A. actinomycetemcomitans* evades disinfection by hydrogen peroxide through this persistence mechanism.

There are some limitations in this study. First, the persistence mechanism is a theoretical assumption and has not been proven. The Lsr family is probably important based on the sequence data in Table 1. The mechanism shown in Figure 5 is a speculation based on the results obtained. Therefore, it is necessary to investigate how the survival of *A. actinomycetemcomitans* as a persister is affected by hydrogen peroxide using gene-knockout or plasmid-overexpressing strains. Next, periodontal disease is not caused by a single bacterial species but by a combination of various pathogenic bacteria [48]. Experiments on a single species of bacteria, as in this study, are essential, but in the future, it will be necessary to verify persister against various species of periodontopathogenic bacteria, such as *Porphyromonas gingivalis* (known as the red complex [49]). In addition, it is important to determine the clinical relevance of the persister. MMC was used for the sterilization of *A. actinomycetemcomitans* persister in this study; however, its relevance and concentration for use in the oral mucosa should be verified. Since MMC is a potent anti-cancer agent [38], it is necessary to search for alternative agents that can be used in the oral cavity, which can sterilize or inhibit an *A. actinomycetemcomitans* persister.

## 5. Conclusions

The effect of hydrogen peroxide disinfection, a conventionally used therapeutic agent, on *A. actinomycetemcomitans*, the causative agent of localized invasive periodontitis, was studied. Of particular importance is the presence of persisters that survive hydrogen peroxide treatment. In the present study, we identified their presence and inferred the putative mechanism of persister formation in detail using RNA sequencing. The Lsr family of transporters is primarily involved in the uptake of AI-2. This inhibition may catalyze the disinfection of periodontal bacteria, suggesting a novel periodontal therapeutic agent. We propose that the persistence of periodontal bacteria discovered in this study cause chronic periodontal disease. The study findings may facilitate the development of new concepts and treatment strategies in dentistry.

## Figures and Tables

**Figure 1 microorganisms-11-01402-f001:**
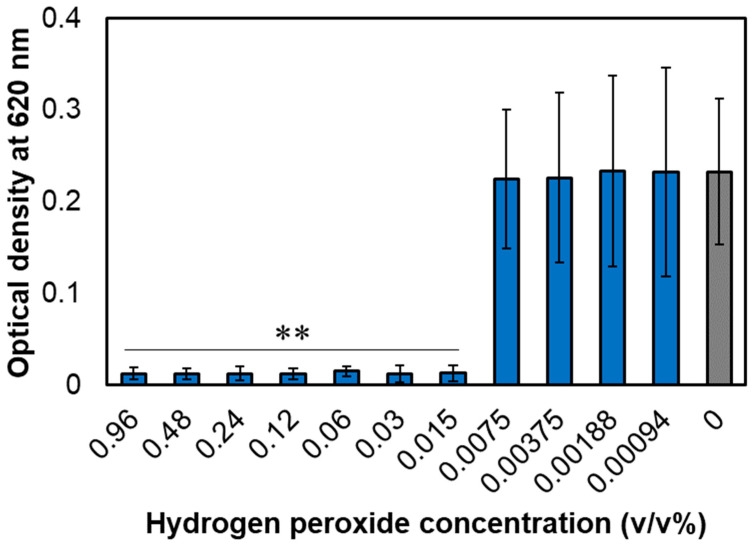
Identification of the minimum inhibitory concentration (MIC) of hydrogen peroxide in BHIY against *A. actinomycetemcomitans*. After incubation for 24 h at different concentrations of hydrogen peroxide (0.96–0% *v/v*; a 2-fold serial dilution was applied), the culture optical densities were measured at an absorbance of 620 nm, which was considered growth. Error bars indicate standard deviations of at least three experiments from each independent culture. Student’s *t*-tests were used to compare the control (0% *v/v*) and other groups (** indicates a *p*-value < 0.01).

**Figure 2 microorganisms-11-01402-f002:**
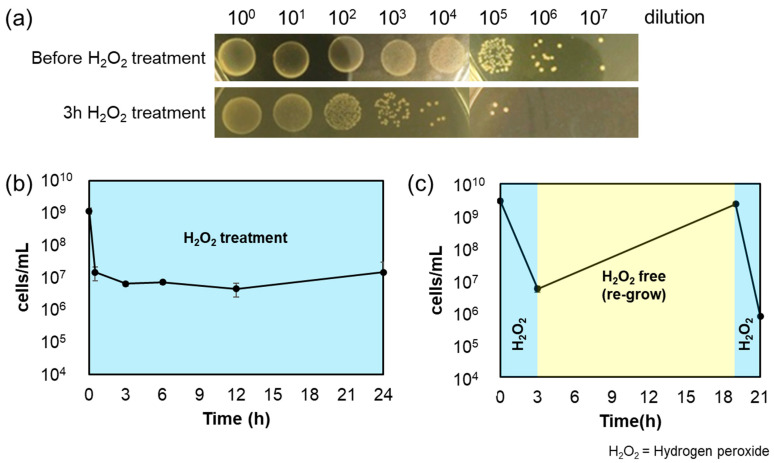
(**a**) Spot-plated colonies of *A. actinomycetemcomitans* before (0 h) and after (3 h) 4× MIC hydrogen peroxide treatment. Each spot was plated on BHIY agar using a 10 µL sample and incubated at 37 °C and 5% CO_2_ for one day. (**b**) Bactericidal effects of 4× MIC hydrogen peroxide against *A. actinomycetemcomitans*. Colonies of the spot-plating at each time (0, 0.5, 3, 6, 12, and 24 h) were counted. (**c**) Confirmation that the *A. actinomycetemcomitans* surviving hydrogen peroxide treatment were persisters and not genetic mutants or antiseptic-resistant bacteria. After hydrogen peroxide treatment for 3 h, the medium was changed to hydrogen peroxide-free BHIY and incubated at 37 °C and 5% CO_2_ for 16 h (yellow). The re-grown culture was again treated with 4× MIC hydrogen peroxide (blue). Error bars indicate standard deviations of at least three experiments from each independent culture.

**Figure 3 microorganisms-11-01402-f003:**
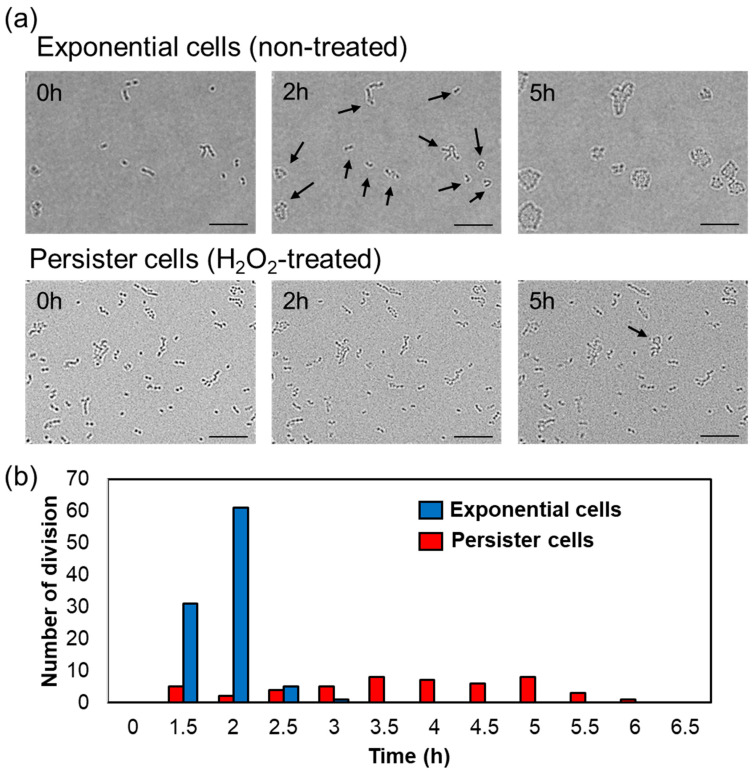
(**a**) Microscopic observation of non-treated exponential cells and hydrogen peroxide-treated persister cells on BHIY gel pads at 0, 2, and 5 h. The scale bar indicates 10 µm. Divided bacteria are indicated by arrows. (**b**) The number of cell divisions of non-treated exponential cells (blue) and hydrogen peroxide-treated persister cells (red) every half an hour.

**Figure 4 microorganisms-11-01402-f004:**
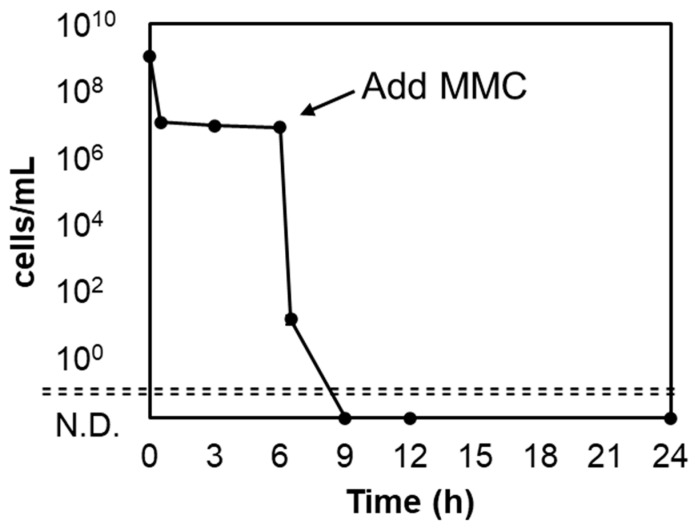
Bactericidal effect of mitomycin C (MMC) on *A. actinomycetemcomitans* persister cells after hydrogen peroxide treatment. Hydrogen peroxide (4× MIC) was added at 0 h and incubated for 6 h. MMC was added after washing with PBS to remove hydrogen peroxide. The number of viable cells was determined by spot-plating for up to 24 h. Error bars indicate standard deviations of at least three experiments from each independent culture. N.D., not detected.

**Figure 5 microorganisms-11-01402-f005:**
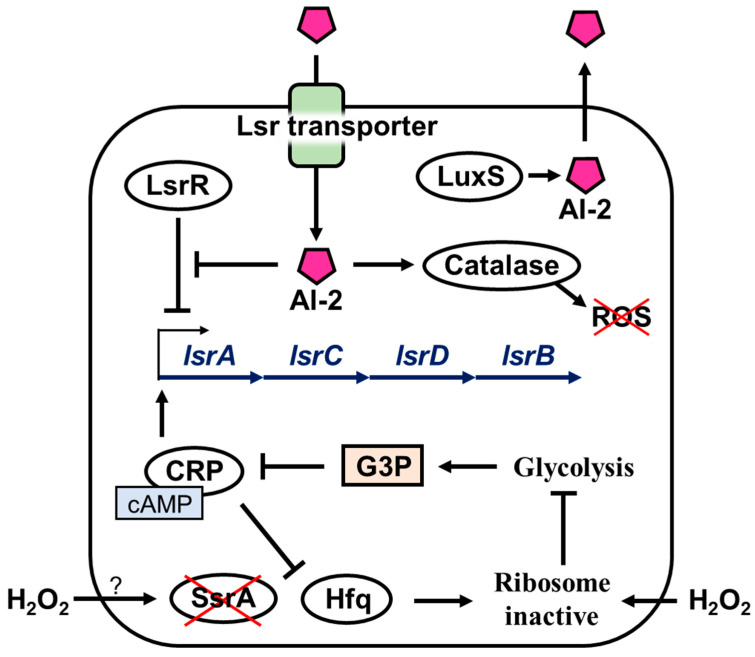
Mechanisms inferred from RNA expression levels. *A. actinomycetemcomitans* exposed to hydrogen peroxide enhances the uptake of AI-2 through upregulation of the Lsr transporter. AI-2 suppresses the function of LsrR, which represses the Lsr family and promotes catalase production. Hydrogen peroxide greatly suppresses the expression of *ssrA*, a ribosome rescue factor, and inactivates ribosomes, although the mechanism is unclear. Metabolism is abolished, glycerol 3-phosphate (G3P) is not produced, and cAMP-CRP further enhances the expression of the Lsr family and the ribosome rescue factor Hfq. → indicates induction, ⟞ indicates repression. × means inhibition or elimination.

**Table 1 microorganisms-11-01402-t001:** Expression levels of key genes of hydrogen peroxide-treated *A. actinomycetemcomitans* using RNA sequencing. Fold change is the relative expression of hydrogen peroxide-treated *A. actinomycetemcomitans* when untreated *A. actinomycetemcomitans* is considered 1. Red indicates increased expression, and blue indicates decreased expression.

Gene	Fold-Change	Description
*lsrA*	2.51	Autoinducer 2 ABC transporter ATP-binding protein
*lsrC*	5.28	Autoinducer 2 ABC transporter permease
*lsrD*	1.56	Autoinducer 2 ABC transporter permease
*lsrB*	0.86	Autoinducer 2 ABC transporter substrate-binding protein
*lsrR*	0.75	Transcriptional regulator
*luxS*	0.95	S-ribosylhomocysteine lyase
HMPREF9996_RS08555	2.52	Catalase
*crp*	0.97	cAMP-activated global transcriptional regulator
*hfq*	0.91	RNA chaperone
*ssrA*	0.31	Transfer-messenger RNA

## Data Availability

The data presented in this article are available upon request from the corresponding author. The sequence data have been deposited with links to BioProject accession number PRJDB15845 in the DDBJ BioProject database.

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
