# Peer review of "Persister Cell Formation and Elevated lsrA and lsrC Gene Expression upon Hydrogen Peroxide Exposure in a Periodontal Pathogen Aggregatibacter actinomycetemcomitans"

_microorganisms, 2023, doi:10.3390/microorganisms11061402_

Round 1

Reviewer 1 Report

The manuscript is well written, and most of the results and discussion are straightforward. Some thoughts and comments to improve the manuscript are provided below:

My major comments are as follows.

1.        P 5, Line 180-181 “A total of … cells were observed.”

Why is the number of observed exponential and persister cells so different? Please tell me if this difference is not a problem in comparing the two cells. Also, how much of each cell was seeded on agarose gel pads? How were the cells observed? (Did you look at the entire gel or just a few random spots and sum them up?) Please let us know those as well.

2.        P 7, Figure 3 (a)

Why are the observation times (2, 6 h) and microscope magnifications presented for each cell different? Please provide 0, 2, and 6-hour images of both groups unless there is a special reason. Also, provide the same microscope magnification.

Here are the minor comments.

P 5, Line 183: Please correct “Figure 3 and Table 2” to “Figure 3 and Table 1”. Then insert the word “Table 2” in the correct position in Results.

Author Response

Thank you for the encouraging feedback. A point by point response has been provided for each query and the corresponding changes in the main manuscript have been mentioned.

  1. P 5, Line 180-181 “A total of … cells were observed.”

Why is the number of observed exponential and persister cells so different? Please tell me if this difference is not a problem in comparing the two cells. Also, how much of each cell was seeded on agarose gel pads? How were the cells observed? (Did you look at the entire gel or just a few random spots and sum them up?) Please let us know those as well.

A. Thank you very much for your helpful comments. When observing bacteria microscopically, it is extremely difficult to determine their viability by appearance. Since most of the exponential cells are viable, we judged that observing approximately 100 cells would be sufficient, and as shown in Figure 3b, almost all of them started dividing within 1.5 to 2 h. In contrast, the hydrogen peroxide-treated bacterial population could not be visually distinguished between dead bacteria and those that survived as persisters; Figure 2b shows that only a small percentage (approximately 0.5 %) were persisters. Hence, it was necessary to evaluate a large number of bacteria. In this experiment, we observed more than 4,000 bacteria, differentiated those that initiated division as persister cells and those that did not as dead bacteria, and measured the division initiation time of these persister cells. As shown in Figure 3b and Table S1 (previously Table 1), the number of evaluated bacteria were sufficient for conclusion, since these evaluated cells showed a clear difference in behavior.

The manuscript lacked a clear description of the experimental method. We thank you for the opportunity to provide clarity here. We have added the following experimental details in paragraph 2.3, page 3, line 113.

“Exponential state (OD600=0.4) and hydrogen peroxide-treated (3 h) bacterial culture of A. actinomycetemcomitans Y4 were centrifuged at 3,500 ×g for 10 min and washed with PBS, respectively. Further, 10 µL of each was placed on the gel pads and observed with a cover glass under a microscope.  All the analysis points were selected randomly, avoiding areas of bacterial aggregation.”

  1. P 7, Figure 3 (a)

Why are the observation times (2, 6 h) and microscope magnifications presented for each cell different? Please provide 0, 2, and 6-hour images of both groups unless there is a special reason. Also, provide the same microscope magnification.

A. Thank you for your valuable suggestion. The microscopic images provided earlier were inconclusive. Hence, as recommended, we have replaced the microscopic images with those of the same time points (0, 2, and 5 h) and magnification, for better comparison (Figure 3a).

Here are the minor comments.

P 5, Line 183: Please correct “Figure 3 and Table 2” to “Figure 3 and Table 1”. Then insert the word “Table 2” in the correct position in Results.

A. We apologize for this oversight. To rectify this, we have transferred the previous Table 1 to supplemental data as Table S1 and Table 2 (currently Table 1) has been referred to in the result section 3.5.

Reviewer 2 Report

Comments to Authors

            In this manuscript, authors showed the existence of A. actinomycetemcomitans persister cells against hydrogen peroxide treatment and the involvement of autoinducer uptake for the persister formation mechanism by RNA sequencing. These findings were fundamentally interesting for understanding the infection mechanism of A. actinomycetemcomitans, the main causative agent of the localized invasive periodontitis. They will show a new concept of a treatment strategy for periodontitis. However, this manuscript has some problems.

Major points

1.     In lines 158-159, the authors concluded that the surviving cells, which re-grew after the 4xMIC hydrogen treatment, were persisters, not genetic mutants, because they were still sensitive to the same concentration of hydrogen oxide. Have the authors checked the mutations for the genes related to the effect of hydrogen oxide, such as superoxide dismutase?

2.     In Figure 2d, the authors tested the hydrogen oxide concentration in the bacterial culture and determined the concentration was above 400 ppm. But the left images of Figure 2d were indistinguishable compared to the standard colorimetric chart. How did they choose the concentration? Please explain the mechanism of this test.

3.     The authors of Lines 206-207 and 256-257 showed that Mitomycin C (MMC) killed the persisters cells. MMC inhibits DNA replication by preventing DNA division and DNA strand breaks caused by reactive oxygen species. If so, why does MMC kill persisters which hardly divide? In addition, they used 10xMIC MMC to kill the persister cells. How did they determine this dose?

Minor points

1.     In Figure 2a, I think the cell concentration/density is in reverse order.

2.     In Line 183, “Figure 3 and Table 2” should be replaced with “Figure 3” because Table 2 shows the fold changes of RNA expression. And Table 2 should be placed after caption 3.5. 

3.     In Figure 3a, the fold of microscope images between Exponential and Persister cells differed. If the authors compared the difference in the cell division rates between these two types of cells, the same fold should be better.

Author Response

 Thank you for reviewing our paper and providing us with all of your helpful comments. We have revised and updated the main text of our manuscript following your suggestions; we have responded to each of your comments in the section below.

Major points

  1. In lines 158-159, the authors concluded that the surviving cells, which re-grew after the 4xMIC hydrogen treatment, were persisters, not genetic mutants, because they were still sensitive to the same concentration of hydrogen oxide. Have the authors checked the mutations for the genes related to the effect of hydrogen oxide, such as superoxide dismutase?

A. We thank the reviewer for the careful reading. In the present study, we have not analyzed genetic mutation. However, as shown in Figure 2c, it is clear that genetic tolerance has not been acquired. It has been specified as a study limitation in the Discussion section that elucidating the detailed genetic mechanism requires the use of knockout strains. In addition, as shown in Table 1 (previously Table 2), the gene expression of catalase, which is involved in reactive oxygen species generation, increased 2.52-fold, which is one of the important factors to avoid hydrogen peroxide sterilization.

  1. In Figure 2d, the authors tested the hydrogen oxide concentration in the bacterial culture and determined the concentration was above 400 ppm. But the left images of Figure 2d were indistinguishable compared to the standard colorimetric chart. How did they choose the concentration? Please explain the mechanism of this test.

A. Thank you for the opportunity to provide clarity here. We apologize for the poor presentation of data. The test described in Figure 2d (currently Figure S1) was performed because there was a possibility that the constant number of viable hydrogen peroxide cells (after approximately 3 h) in the experiment in Figure 2b might be due to a decrease in hydrogen peroxide concentration. As shown in the revised Figure S1 (b), the hydrogen peroxide concentration can be confirmed by the color change. Figure S1 (a) shows the hydrogen peroxide concentration during the experiment described in Figure 2b, which is ≥ 0.06 % concentration in Figure S1 (b), because 0.06 % hydrogen peroxide was added to the A. actinomycetemcomitans culture in Figure 2b. If the concentration had decreased during the 24-h hydrogen peroxide treatment, the color of the test paper would be lighter. However, as shown in Figure S1 (a), the colors are similar at 24 h, indicating that the hydrogen peroxide concentration in the culture was maintained. Therefore, the constant number of viable bacteria shown in Figure 2b is not due to a decrease in hydrogen peroxide concentration, but due to survival of the bacteria as persisters.

For clarity, we have revised the legend for Figure S1 and provided the following revised description on page 5, line 181 of the main manuscript: “In the experiment shown in Figure 2b, the hydrogen peroxide concentration in the A. actinomycetemcomitans culture was determined at 0, 0.5, 12, and 24 h, and no decrease in concentration was observed (Figure S1a). Bacterial culture with 0.06 % hydrogen peroxide showed concentrations ≥ 0.06 % at 0 h, and the color of the test paper did not change after 24 h (Figure S1a-b).”

  1. The authors of Lines 206-207 and 256-257 showed that Mitomycin C (MMC) killed the persisters cells. MMC inhibits DNA replication by preventing DNA division and DNA strand breaks caused by reactive oxygen species. If so, why does MMC kill persisters which hardly divide? In addition, they used 10xMIC MMC to kill the persister cells. How did they determine this dose?

A. Thank you for your pertinent query. The mechanism of action of MMC for killing bacterial persisters has already been reported in previous papers (PMID: 25858802). MMC is passively transported and bioreductively activated to cause the spontaneous cross-linking of DNA. This has been observed in both exponential and persister cells. In the paper mentioned above, the effect of MMC disinfection on Escherichia coli persister was verified at concentrations of 5–10× MIC. With this range as a reference, we tested MMC at a concentration of 10× MIC to ensure that the MIC is sufficiently compared to the MIC for A. actinomycetemcomitans. This following information has been included in the revised manuscript:

page 6, line 215: “, the sterilization of A. actinomycetemcomitans persister cells was confirmed at 10× MIC of MMC (1.25 µg/mL). The MIC data of MMC against A. actinomycetemcomitans is shown in Figure S2. As a result, significant and rapid sterilization was achieved (Figure 4), demonstrating the efficacy of MMC against A. actinomycetemcomitans persisters.”

page 9 line 284: “MMC is passively transported inside bacterium and is bioreductively activated, causing spontaneous cross-linking of DNA; MMC activity does not require active metabolism, making it effective against persister cells.”

Minor points

  1. In Figure 2a, I think the cell concentration/density is in reverse order.

A. We apologize for the typographical error. We have revised the axes legends for the graph (Figure 2a).

  1. In Line 183, “Figure 3 and Table 2” should be replaced with “Figure 3” because Table 2 shows the fold changes of RNA expression. And Table 2 should be placed after caption 3.5. 

A. We thank the reviewer for the careful reading. As suggested, we have placed Table 2 (currently Table 1) under heading 3.5. The previous Table 1 has been moved to supporting information as Table S1.

  1. In Figure 3a, the fold of microscope images between Exponential and Persister cells differed. If the authors compared the difference in the cell division rates between these two types of cells, the same fold should be better.

A. As suggested, we have replaced the microscopic images with those of the same time points (0, 2, and 5 h) and magnification, for better comparison (Figure 3a).

Reviewer 3 Report

1. What are the future continuations of this study?

2. What are the limitations of this study?

3. What is the age range of patients most affected?

1. What are the future continuations of this study?

2. What are the limitations of this study?

3. What is the age range of patients most affected?

Author Response

Thank you for reviewing our paper and providing us with all of your helpful comments. We have revised and updated the main text of our manuscript following your suggestions; we have responded to each of your comments in the section below.

  1. What are the future continuations of this study?

A. The mechanism of persister shown in Figure 5 is only an inference based on the results, and we believe that the mechanism needs to be supported using knockout strains and other methods in the future. Furthermore, since this study only shows the phenomenon for the combination of A. actinomycetemcomitans and hydrogen peroxide, we intend to examine the risk of persister using other periodontal bacteria (red complex bacteria, such as Porphyromonas gingivalis) and evaluate other disninfectants in future. The scope for future research has been included in the discussion section of the manuscript.

Page 10, line 339: “The mechanism shown in Figure 5 is a speculation based on the results obtained. Therefore, it is necessary to investigate how the survival of A. actinomycetemcomitans as a persister is affected by hydrogen peroxide using gene-knockout or plasmid-overexpressing strains. Next, periodontal disease is not caused by a single bacterial species, but a combination of various pathogenic bacteria [48]. Experiments on a single species of bacteria, as in this study, are essential, but in the future it will be necessary to verify persister against various species of periodontopathogenic bacteria, such as Porphyromonas gingivalis (known as the red complex [49]). In addition, it is important to determine the clinical applicability of the persister. MMC was used for the sterilization of A. actinomycetemcomitans persister in this study; however, its applicability and concetration for use in the oral mucosa should be verified. Since MMC is a potent anticancer agent [37], it is necessary to search for alternative agents that can be used in the oral cavity, which can sterilize or inhibit A. actinomycetemcomitans persister."

  1. What are the limitations of this study?

A. Thank you for your insightful query. There are some limitations to the study. First, as clarified earlier, the mechanism of persister is a theoretical assumption, which has not been proved experimentally. It is necessary to investigate the survival mechanism of a persister using strategies such as gene-deficient strains or gene-overexpressing strains using plasmids. Second, periodontal disease is not caused by a single bacterial species, but is a combination of various pathogenic bacteria. Experiments using a single bacterial species, as in this study, are essential; however, in future it will be necessary to verify the effect of persister on various species of periodontopathic bacteria. In addition, it is important to determine the clinical applicability of the persister. MMC was used for the sterilization of A. actinomycetemcomitans persister in this study; however its applicability and optimum concentration for use in the oral mucosa must be verified. Since MMC is a potent anticancer agent, it is also necessary to identify alternative agents that can be used in the oral cavity and that can sterilize or inhibit A. actinomycetemcomitans persister. In view of your comment, the limitations have been included in the discussion section, page 10, line 339.

  1. What is the age range of patients most affected?

A. Previously, the term "juvenile periodontitis" was used to describe the disease, which mainly affected younger patients aged below 35 years of age. However, recent revisions have summarized it as "rapidly progressing periodontitis", indicating that it is associated with a wide range of age groups (PMID: 29926951). Therefore, this study is applicable to all age groups with periodontitis. In view of your comment, we have added this information on page 1, line 31.

Reviewer 4 Report

The Authors studied (1) the viability of a model strain bacteria on a solid medium after hydrogen peroxide exposure; (2) Persister cells resuscitation time on agarose gel pads in light microscopy experiment; (3) an effect of mitomycin C on the presence of persisters in hydrogen peroxide-treated bacterial population; (4) Transcriptome profile of hydrogen peroxide treated bacteria, with the inclusion of untreated control.

I am concerned with insufficient experimental evidence provided in the manuscript. (1) Most experiments were done only once, whereas biological replicates are a standard. Particularly, the experiments on hydrogen peroxide MIC determination, determination of bactericidal effects of 4x MIC hydrogen peroxide against A. actinomycetemcomitans and persister resuscitation should be repeated in triplicates. (2) The role of lsrA and lsrC gene products in persister survival was not elucidated properly. Neither gene knockout experiment nor autoinducer uptake experiment were performed. Please see PMID: 15601708 for an example. (3) Nucleotide sequence data generated during the study must be deposited to one of the INSDC databases. For example, one can use the website https://www.ncbi.nlm.nih.gov/bioproject/.

Other considerations:

Find in the text and remove the word “drug” in the relation to hydrogen peroxide. It is not a drug as such, but a mild antiseptic with a potential to induce human cell death and apoptosis (PMID: 16753840; PMID: 36585955).

Line 30. The ability to form something cannot be considered a trigger. Please revise.

Line 35. “was also focused”. The primary focus of the study cannot be understood from the previous text.

Line 76. change target bacterium to model strain

Line 82 change dilution to dilutions, if there were a number of them.

Lines 84, 92. turbidity —> optical density

Section 2.1 please disclose the range of tested hydrogen peroxide concentrations

Line 97. Bacterial solution —> bacterial suspension

Line 100. In the current context, drug-resistant is a confusing term. Please find a substitution.

Lines 118-119. Please provide the reference for antibacterial MIC or describe how it was obtained with the references to employed testing procedures.

Lines 137-138. As follows from Figure 1, 0.015% hydrogen peroxide inhibited the bacterial growth. Change the sentence to something like “Hydrogen peroxide inhibited A. actinomycetemcomitans growth at concentration 0.015% and above”.

Line 141. Please clarify: “in brain heart infusion broth with yeast extract”.

Figure 1. Vertical axis is interpreted as growth inhibition, but actually shows optical density values. Please correct.

Figure 2d can be removed, since it does not contain important information.

Line 150 change the bacteria culture to the bacterial culture

Lines 162, 163. Calculate percentage values equal to given parts per million values and use in the text together.

Line 177. Please clarify how “elongation” is connected to “resuscitation”. What is the meaning behind “elongation”?

Table 1 does not contain additional information in comparison to Figure 3b and should be removed.

Line 219. The elevated expression of lsrA and lsrC genes does not provide a proof of their role in hydrogen peroxide exposure survival. A claim of this type must be supported by a gene knockout experiment.

 Moderate English editing is required.

Author Response

Thank you for reviewing our paper and providing us with all of your helpful comments. We have revised and updated the main text of our manuscript following your suggestions; we have responded to each of your comments in the section below.

(1) Most experiments were done only once, whereas biological replicates are a standard. Particularly, the experiments on hydrogen peroxide MIC determination, determination of bactericidal effects of 4x MIC hydrogen peroxide against A. actinomycetemcomitans and persister resuscitation should be repeated in triplicates.

A. Thank you very much for your helpful comments. All experiments have been performed with a minimum of three independent cultures, which has been indicated in the main text. For better clarity, we have revised the phrasing to indicate biological replication.

(2) The role of lsrA and lsrC gene products in persister survival was not elucidated properly. Neither gene knockout experiment nor autoinducer uptake experiment were performed. Please see PMID: 15601708 for an example. 

A. We apologize if the data conveyed such a remark. As recommended by you, confirmatory experiments such as those using gene-deficient strains are needed to corroborate the RNA sequencing data shown in Table 1 (previously Table 2) and the mechanism shown in Figure 5. However, in this was not addressed in the present study and has been indicated as a limitation of the present study and a scope for future research (page 10, line 339). Instead, we have added data on the expression levels of lsrA and lsrC obtained using RNA sequencing and confirmed these data using real-time RT-PCR (Figure S3).

(3) Nucleotide sequence data generated during the study must be deposited to one of the INSDC databases. For example, one can use the website https://www.ncbi.nlm.nih.gov/bioproject/.

A. We thank the reviewer for the valuable suggestion. We have deposited our sequence data on the databases (BioProject accession: PRJDB15845). This has also been indicated in the “Data Availability Statement” of the main manuscript.

Other considerations:

Find in the text and remove the word “drug” in the relation to hydrogen peroxide. It is not a drug as such, but a mild antiseptic with a potential to induce human cell death and apoptosis (PMID: 16753840; PMID: 36585955).

A. We apologize for the misinterpretation. In view of your comment, we have revised all instances in the manuscript mentioning the term “drug” in relation to hydrogen peroxide.

Line 30. The ability to form something cannot be considered a trigger. Please revise.

A. As suggested, we have revised the term to “induced”.

Line 35. “was also focused”. The primary focus of the study cannot be understood from the previous text.

A. As suggested by you, we have revised the term “focused” to “used”.

Line 76. change target bacterium to model strain

A. This has been revised, per your suggestion.

Line 82 change dilution to dilutions, if there were a number of them.

A. This has been revised, per your suggestion.

Lines 84, 92. turbidity —> optical density

A. This has been revised, per your suggestion.

Section 2.1 please disclose the range of tested hydrogen peroxide concentrations

A. Thank you for your comment. We have indicated this at section 2.2 (line 107).

Line 97. Bacterial solution —> bacterial suspension

A. We have revised the term, per your suggestion.

Line 100. In the current context, drug-resistant is a confusing term. Please find a substitution.

A. Thank you for pointing it out. We have revised it to “antiseptic-resistant” as you suggested.

Lines 118-119. Please provide the reference for antibacterial MIC or describe how it was obtained with the references to employed testing procedures.

A. As requested, we have provided the MIC data as supplemental information in Figure S2.

Lines 137-138. As follows from Figure 1, 0.015% hydrogen peroxide inhibited the bacterial growth. Change the sentence to something like “Hydrogen peroxide inhibited A. actinomycetemcomitans growth at concentration 0.015% and above”.

A. Thank you for your suggestion. We have revised this sentence accordingly.

Line 141. Please clarify: “in brain heart infusion broth with yeast extract”.

A. Brain heart infusion broth with yeast extract is the culture medium used, which has been abbreviated as BHIY in the manuscript. We added “BHIY” in Figure 1 legend (line 159).

Figure 1. Vertical axis is interpreted as growth inhibition, but actually shows optical density values. Please correct.

A. We apologize for the error here. We have revised the vertical axis of Figure 1 and its legend.

Figure 2d can be removed, since it does not contain important information.

A. Thank you for your suggestion. We wished to illustrate that the hydrogen peroxide concentration did not reduce in the experiment described in Figure 2b. Hence, Figure 2d was slightly modified and move to Supporting information as Figure S1.

Line 150 change the bacteria culture to the bacterial culture

A. We have made the required revision.

Lines 162, 163. Calculate percentage values equal to given parts per million values and use in the text together.

A. Thank you for highlighting this. We have revised all the “ppm” values to “%” in the main text.

Line 177. Please clarify how “elongation” is connected to “resuscitation”. What is the meaning behind “elongation”?

A. For clarity, we have removed the term “elongation” since no elongated cells were observed. In a previous study, division or elongation of E. coli persister cells has been defined as resuscitation (PMID: 31926430). Therefore, we had indicated both division and elongation. Thank you for your valuable suggestion.

Table 1 does not contain additional information in comparison to Figure 3b and should be removed.

A. We assumed that it was necessary to show the numerical data for Figure 3b. However, as mentioned by you, it does not contain enough additional information to be included in the text. Hence, we have moved this to Supporting information as Table S1.

Line 219. The elevated expression of lsrA and lsrC genes does not provide a proof of their role in hydrogen peroxide exposure survival. A claim of this type must be supported by a gene knockout experiment.

A. We appreciate this insightful comment. As noted above, the present study does not verify the involvement of the lsr family in A. actinomycetemcomitans persister. We completely agree with your suggestion that future experiments with mutated strains are necessary. This has been indicated in the scope for future research.

This paper was performed English language editing by Editage (www.editage.com).

Round 2

Reviewer 1 Report

This manuscript has been revised well. I am fully satisfied with the authors' response.

Author Response

Thank you very much for providing important comments. We are thankful for the time and energy you expended.

Reviewer 2 Report

The rivsed mamuscript is worhty of publication.

Author Response

(The authors gave the same response as above.)

Reviewer 4 Report

The Authors revised the manuscript and replied the comments in a satisfactory way. However, there is a need for some polishing corrections. The Line numbers here refer to the plain version of the revised Manuscript.

First of all, the Title and Abstract should be brought into agreement with the rest of the article. The current title stands as “Autoinducer uptake by the Lsr transporter triggers Aggregatibacter actinomycetemcomitans persister cells after hydrogen peroxide treatment”. I am concerned that no autoinducer uptake experiments have been performed. The only evidence available from the study is related to lsrA and lsrC gene expression. The persistence mechanism based on this possible uptake is a second-level theoretical assumption, which has not been proven. A better title for the publication may be “Persister cell formation and elevated lsrA and lsrC gene expression upon hydrogen peroxide exposure in a periodontal pathogen Aggregatibacter actinomycetemcomitans”, if you want. Please argue if you believe there is more precise formulation.

In the Abstract, the final statement refers to treatment failure. But, the MS does not contain a clinical or animal model experiment part. Moreover, in the Introduction, I did not find references to literature describing problems associated with the medical use of hydrogen peroxide. Please resolve these contradictions.

In "peroxide-resistance" and "antiseptic-resistance", substitute spaces for hyphens.

Line 17. 0.5 % of the population → 0.5 % of the bacterial population

Lines 20-21. “This study infers that persisters are formed by autoinducer uptake”. The study does not do that.

Lines 34-36. The term “immunodominance” refers to a phenomenon related to lymphocytes (PMID: 11926408 PMID: 22391152 PMID: 34534732). Please note that your Ref. 4 does not use it. I recommend to revise the sentence as “Seven serotypes of A. actinomycetemcomitans have been identified based on surface carbohydrate antigens”.

Line 41. Prototype strain → model strain.

Line 41. “This strain also affects (...) infections.” Please revise.

Line 47. “destroying the pathogen with drug treatment”. Revise.

Line 50. A space is lost. Also, two “exposure”-s in one sentence.

Line 54. Lost space.

Lines 54-55. Revise the sentence.

Lines 57-59. The provided information is insufficient to conclude that “persistent periodontal disease (...) and other oral diseases are difficult to cure completely because persisters (...). I suspect, there has to be a contribution of true drug resistance, based on mutational mechanisms. If the statement is correct, indeed, extend the section and cite more literature.

Line 70. A supporting citation is needed.

Line 72. “the causative agent of localized invasive periodontitis”. Please remove, you already mentioned it in the first paragraph of the Introduction.

Line 77. Is there a better section title?

Line 91. “of independent culture.” Remove.

Lines 167-169. Please clarify that the bacteria were spot-plated after the incubation in the presence of HP. For that purpose I would merge the two sentences.

Line 205. “The cells were divided”. The cells divided? Or something else?

Line 348. applicability → relevance.

Figure 4. The use of Student's t-test for small samples implies the normal distribution of the data. Did you confirm the normality? Otherwise, please use non-parametric test. In the caption, please provide the full spelling of MMC.

Lines 367-368. inferred the mechanism → inferred the putative mechanism

Please do read the whole paper once again and think what can be improved.

Author Response

The Authors revised the manuscript and replied the comments in a satisfactory way. However, there is a need for some polishing corrections. The Line numbers here refer to the plain version of the revised Manuscript.

Response: We really appreciate all of the reviewer’s helpful and constructive comments. We have revised and updated the main text of our manuscript following the reviewer’s suggestions and our point-by-point responses to their comments are provided below.

First of all, the Title and Abstract should be brought into agreement with the rest of the article. The current title stands as “Autoinducer uptake by the Lsr transporter triggers Aggregatibacter actinomycetemcomitans persister cells after hydrogen peroxide treatment”. I am concerned that no autoinducer uptake experiments have been performed. The only evidence available from the study is related to lsrA and lsrC gene expression. The persistence mechanism based on this possible uptake is a second-level theoretical assumption, which has not been proven. A better title for the publication may be “Persister cell formation and elevated lsrA and lsrC gene expression upon hydrogen peroxide exposure in a periodontal pathogen Aggregatibacter actinomycetemcomitans”, if you want. Please argue if you believe there is more precise formulation.

Response: We thank the reviewer for the comment. As pointed out by the reviewer, our paper did not prove the effect of the Lsr transporter on persister. Thus, we have revised the title of our manuscript as suggested.

In the Abstract, the final statement refers to treatment failure. But, the MS does not contain a clinical or animal model experiment part. Moreover, in the Introduction, I did not find references to literature describing problems associated with the medical use of hydrogen peroxide. Please resolve these contradictions.

Response: We thank the reviewer for pointing this out. We have revised the Abstract.

In "peroxide-resistance" and "antiseptic-resistance", substitute spaces for hyphens.

Response: As suggested by the reviewer, we have revised the text.

Line 17. 0.5 % of the population → 0.5 % of the bacterial population

Response: As pointed out by the reviewer, we have added “bacterial” in line 17.

Lines 20-21. “This study infers that persisters are formed by autoinducer uptake”. The study does not do that.

Response: We thank the reviewer for pointing it out. We have revised the abstract.

Lines 34-36. The term “immunodominance” refers to a phenomenon related to lymphocytes (PMID: 11926408 PMID: 22391152 PMID: 34534732). Please note that your Ref. 4 does not use it. I recommend to revise the sentence as “Seven serotypes of A. actinomycetemcomitans have been identified based on surface carbohydrate antigens”.

Response: We thank the reviewer for pointing it out. We have revised the sentence as suggested (line 33).

Line 41. Prototype strain → model strain.

Response: As suggested, we have revised the text (line 39).

Line 41. “This strain also affects (...) infections.” Please revise.

Response: As pointed out, we have revised the sentence (line 39).

Line 47. “destroying the pathogen with drug treatment”. Revise.

Response: As pointed out, we have revised the sentence (line 45).

Line 50. A space is lost. Also, two “exposure”-s in one sentence.

Response: As pointed out by the reviewer, we have added the space and removed one “exposure” (line 49).

Line 54. Lost space.

Response: We thank the reviewer for pointing this out. We have added the space (line 49).

Lines 54-55. Revise the sentence.

Response: As pointed out, we have revised the sentence (line 50).

Lines 57-59. The provided information is insufficient to conclude that “persistent periodontal disease (...) and other oral diseases are difficult to cure completely because persisters (...). I suspect, there has to be a contribution of true drug resistance, based on mutational mechanisms. If the statement is correct, indeed, extend the section and cite more literature.

Response: As mentioned by the reviewer, this sentence was nuanced to mean that "only persister contributes." We agree with the reviewer that drug-resistant bacteria also have an impact on drug treatment. However, what we intend to relay here is the several times a patient is cured and relapses. We believe that the persister is a major factor in the survival of the bacteria. In contrast, for drug-resistant bacteria, after relapse, the same drug does not lead to cure. We wanted to express this difference. We have revised the text again based on this reference (line 53).

Line 70. A supporting citation is needed.

Response: As suggested, we have added the appropriate citation (line 70).

Line 72. “the causative agent of localized invasive periodontitis”. Please remove, you already mentioned it in the first paragraph of the Introduction.

Response” As pointed out, we have removed the expression (line 74).

Line 77. Is there a better section title?

Response: We appreciate the reviewer’s comment. We have revised section 2.1 title.

Line 91. “of independent culture.” Remove.

Response: As suggested, we have removed the expression (line 94).

Lines 167-169. Please clarify that the bacteria were spot-plated after the incubation in the presence of HP. For that purpose I would merge the two sentences.

Response: As suggested, we have merged the two sentences to clarify the bactericidal condition (line 170).

Line 205. “The cells were divided”. The cells divided? Or something else?

Response: As pointed out by the reviewer, we have revised the text (line 208).

Line 348. applicability → relevance.

Response: As suggested by the reviewer, we have revised the terminology (line 357, 358).

Figure 4. The use of Student's t-test for small samples implies the normal distribution of the data. Did you confirm the normality? Otherwise, please use non-parametric test. In the caption, please provide the full spelling of MMC.

Respose: We thank the reviewer for pointing this out. We used a non-parametric test as suggested by the reviewer. However, non-significance was obtained from both Mann–Whitney U test and Wilcoxon rank sum test. Hence, we deleted the sentence about the test. Although there was no significant difference between 6 and 6.5 h of Figure 4, the number of viable bacteria reduced to zero after at least 3 hours of MMC treatment. Hence, this finding shows that MMC can completely sterilize A. actinomycetemcomitans persister. We have revised the explanation in the text (line 226), as well as added the full spelling of MMC in figure 4 caption.

Lines 367-368. inferred the mechanism → inferred the putative mechanism

Response: As suggested, we have revised the expression (line 376).